# Effects of Extraction Technique on the Content and Antioxidant Activity of Flavonoids from Gossypium *Hirsutum linn*. Flowers

**DOI:** 10.3390/molecules27175627

**Published:** 2022-08-31

**Authors:** Jiaxing Dong, Kehai Zhou, Xiaoyang Ge, Na Xu, Xiao Wang, Qing He, Chenxu Zhang, Jun Chu, Qinglin Li

**Affiliations:** 1Key Laboratory of Xi’an Medicine, Ministry of Education, Anhui University of Chinese Medicine, Hefei 230038, China; 2Research and Technology Center, Anhui University of Chinese Medicine, Hefei 230038, China; 3State Key Laboratory of Cotton Biology, Institute of Cotton Research, Chinese Academy of Agriculture Sciences (CAAS), Anyang 455000, China; 4State Key Laboratory of Tea Plant Biology and Utilization, School of Tea & Food Science and International Joint Laboratory on Tea Chemistry and Health Effects of Ministry of Education, Anhui Agricultural University, Hefei 230036, China

**Keywords:** cotton flowers flavonoids, extraction process, response surface methodology, antioxidant activity

## Abstract

Cotton is one of the Uyghur medical materials in China and is rich in flavonoids. Flavonoids have important pharmacological effects. The yield of flavonoids in traditional extraction methods is low, which affects the development of flavonoids. Therefore, it is urgent to optimize the extraction techniques. The yield of flavonoids in cotton flowers was effectively improved by response surface methodology, and the highest yield of flavonoids reached 5.66%, and the optimal extraction process conditions were obtained. The DPPH free radical scavenging rate, hydroxyl free radical scavenging rate, superoxide anion free radical scavenging rate, and reducing ability were tested to reflect the antioxidant capacity of flavonoids. The flavonoids had an excellent antioxidant effect. Cell experiments suggested that the flavonoids had the effect of protecting glutamate-induced damage to HT-22 cells. The results of this study provide a theoretical basis for the extraction of cotton flowers flavonoids and the comprehensive evaluation of antioxidant products, as well as the extraction of other plant flavonoids.

## 1. Introduction

Cotton is one of the most significant crops globally, and it has a long history of industrial application. In addition, cotton is widely grown in China, the United States, India, and other countries [1]. Xinjiang cotton is the most popular because of its best quality in China [2]. The utilization of cotton and its by-products will bring huge economic benefits to agricultural development. Moreover, cotton flowers contain antioxidant flavonoids, which significantly affect several neurological diseases such as Alzheimer’s Disease (AD) and Parkinson’s Disease (PD), according to Xinjiang Uygur medicine [3].

The flavonoids in cotton flowers have pharmacological effects such as sedation, brain strengthening, and heart-strengthening [4]; the antioxidant effects of free radicals contribute to the pharmacological activity of flavonoids. Therefore, it is imperative to efficiently extract flavonoids from cotton flowers. However, at present, in conventional extraction methods, such as solvent extraction or hot water extraction, the process flow of these extraction methods is very complicated, with a long extraction time, insufficient extraction of active ingredients, and a low extraction rate [5].

RSM is a mathematical and statistical optimization analysis method that can accurately explain the link between response values and numerous parameters [6,7]. RSM was developed to optimize the extraction process of flavonoids [8,9,10]. The current extraction technology is mainly applied to other plants except for cotton (Xinjiang jujube [11], navel orange [12], and rape bee pollen [13]). Extracting flavonoids from discarded cotton flowers has great economic value. For this reason, our research group developed a RSM to extract flavonoids.

Flavonoids have an antioxidant activity such as ·OH Free Radical Scavenging Ability, DPPH%-UV radical scavenging assay [14,15,16], but few studies deal with the antioxidant activity of cotton flowers flavonoids in vitro, especially in Hydroxyl radical scavenging test, Superoxide anion radical scavenging activity. Current work focused on the following five aspects: DPPH radical scavenging test, hydroxyl radical scavenging test, free radical scavenging rate of superoxide anion, Fe^3+^ reduction ability, and the protective effect of cotton flowers flavonoids on mouse hippocampal neuronal cells HT-22, comprehensively evaluating the antioxidant activity and cell protection effect of cotton flowers flavonoids.

## 2. Materials and Methods

### 2.1. Drugs and Reagents

Cotton flowers came from the CAAS Cotton Research Institute, Wang Jiang city planting base, Anhui Province. Cotton flowers were divided into two parts by different pretreatment methods: one was treated with liquid nitrogen (LNT) and ground into powder. The other treatment was dried for 3 h at 60 °C before being ground into powder (DT). All powder was stored in a −80 °C freezer before the further experiment. Anhydrous ethanol, 70% ethanol, DPPH (Synonyms:2,2-Diphenyl-1-picrylhydrazyl), Aluminum nitrate, Sodium hydroxide, Hydrogen peroxide, Trichloroacetic acid, Ferrous sulfate, Phosphoric acid buffer, and Salicylic acid, were all from Sinopharm Group.

### 2.2. Ultraviolet Spectroscopy

The reference substance and the extract were scanned at 200–500 nm. Rutin standard was selected to compare the cotton flower flavonoids, and a standard curve was developed to determine the concentration of cotton flower flavonoids.

### 2.3. RSM Optimization Test Design

On the premise that the optimal scheme of four factors was obtained according to the single factor test results, the optimal scheme was set as 0, and the data ± 5 under the premise of the optimal scheme was set as −1, 1, respectively. The selected factor levels and codes are shown in Appendix A.

### 2.4. Extraction Steps

The first step is soaking; the cotton flowers powder is put into ethanol. The second step is extraction. The third step is centrifugation. The flavonoids can be obtained by final purification and freeze-drying.

### 2.5. Preparation of Cotton Flower Flavonoid Solution

0.04 g of lyophilized cotton flowers flavonoids powder was weighed, dissolved in distilled water, and transferred to a 100 mL flask for sample preparation, dilute with distilled water into flavonoids solutions of 0.2, 0.1, 0.05, and 0.025 mg/mL.

### 2.6. DPPH Radical Scavenging Test

The DPPH solution was made in the following manner [17]: 0.00788 g DPPH was weighed. It was dissolved with anhydrous ethanol and then transferred to a 20 mL flask, and stored at −20 °C for later use. Before use, 2 mL of DPPH solution was pipetted and diluted to 10 mL with 100% anhydrous ethanol. According to the methods in the published literature [18,19], the DPPH free radical scavenging activity of cotton flowers flavonoids was determined by spectrophotometry. The specific methods were as follows:A11.0 mL of flavonoid sample solution was placed into a 10 mL tube for centrifugation, 2.0 mL DPPH solution was reacted in the dark for 30 min at room temperature (RT). At 517 nm, the absorbance of the solution was calculated, and the blank solution was 100% anhydrous ethanol;A21.0 mL of flavonoid sample solution of different concentrations was placed in a centrifuge tube, 2.0 mL of 100% anhydrous ethanol was added, mixed, and the absorbance of the solution was measured at 517 nm;A3the light absorption of the solution was measured at 517 nm after 1.0 mL of 100% anhydrous ethanol was added to a 2.0 mL DPPH solution. To test the antioxidant activity of flavonoids in cotton flowers, blank handling treatment (BHT) was utilized as a positive control group. DPPH free radical scavenging activity calculation:

DPPH free radical scavenging activity (%) = [1 − (A1 − A2)/A3] × 100%.

### 2.7. Hydroxyl Radical Scavenging Test

According to the methods in recent literature [20,21], the hydroxyl radical scavenging test steps are as follows: 1 mL of cotton flowers flavonoid solution of different concentrations was put into a 10 mL test tube, 1 mL of 6 mmol/L FeSO_4_ was added then, followed by 1 mL of 6 mmol/L H_2_O_2_. After 10 min, 1 mL of 6 mmol/L ethanol-salicylic acid solution was mixed with the previous mixed solution at 37 °C for 30 min. Finally, at 510 nm, the absorbance (Aa) was measured. The control group (Ab) replace the salicylic acid with distilled water. The blank group (Ac) replace the flavonoid solution with distilled water. As a positive control group, BHT was used. The hydroxyl radical scavenging activity was determined using the formula below: OH radical scavenging rate = [1 − (Aa − Ab) / Ac] × 100%.

### 2.8. Superoxide Anion Radical Scavenging Activity

The method of Barrientos, C. [22,23] enhanced superoxide anion free radical scavenging activity. 4.5 mL of 50 mmol/L Tris-HCl buffer solution (pH 8.2) was put in the test tube, and then 1 mL of cotton flavonoid sample solutions of different concentrations were added and placed in a 25 °C water bath for 20 min. At the same time, the prepared 25 mmol/L pyrogallic acid was also heated in a water bath at 25 °C for 20 min. Then, the sample tube was filled with 0.4 mL of preheated pyrogallic acid solution and reacted at 25 °C for 4 min. Finally, 1 mL of 10 mmol/L HCl was added to stop the reaction. A1 represents the absorbance value recorded at 325 nm wavelength.

The cotton flower flavonoid solution was replaced with distilled water, which is the control group, which is recorded as A0.

4.5 mL 50 mmol/L Tris-HCl buffer solution (pH 8.2) was added to 1 mL cotton flower flavonoid solution of different concentrations and placed at 25 °C for 20 min. Instead of pyrogallic acid, the test tube was filled with 10 mmol/L HCl. which is recorded as A2. As a positive control, BHT was used. The rate of scavenging superoxide anion radicals was estimated as follows:Radical scavenging action of superoxide anion (%) = [1 − (A1 − A0) / A2] × 100%.

### 2.9. Determination of Reducing Capacity

2 mL of flavonoid solutions of different concentrations and 2 mL of phosphate buffer (NaH_2_PO_4_ and Na_2_HPO_4_, 0.2 mol/L, pH 6.6) were mixed, 2 mL of 1% potassium ferricyanide solution was added, and the solution was soaked at 50 °C after thorough mixing in the water for 20 min. After being cooled, 2 mL of trichloroacetic acid (10%) was added to stop the reaction. The solution was centrifuged for 20 min at 3000× *g* at RT. 2 mL supernatant and 0.5 mL ferric chloride solution (1%) were extracted. The light absorption value was measured at 700 nm.

### 2.10. Cell Culture

In a humidified cell incubator, the cells were cultured at 37 °C and 5% CO_2_ in a Dulbecco’s modified eagle medium (DMEM) mixture containing 10% fetal bovine serum and 1% penicillin streptomycin. DMEM and antibiotic were purchased from Gibco. The HT-22 cells were collected and washed three times in Phosphate Buffered Saline (PBS) with a pH of 7.4. Then the cells were prepared into a 5% cell suspension with PBS [24,25]. HT-22 cells were purchased from Shanghai Saiqi Biological (Shanghai, China).

### 2.11. MTT Test

The effect of flavonoid on HT-22 cell viability was determined by MTT assay. HT-22 cells (1 × 10^5^ cells/well) were treated with varying concentrations (1, 5, 25, 100 μg/mL) of flavonoid in 96-well plates for 24, 48, or 72 h in DMEM. During the last 4-h incubation, cells were treated with MTT (0.5 mg/mL), followed by adding DMSO to dissolve crystallization, then measuring the absorbance at 490 nm.

### 2.12. Statistical Analysis

GraphPad Prism 5 (San Diego, CA, USA) was used for the analysis, and Design Expert 8.0.6 was used to conduct the statistical analysis. The data was examined using the t-test or one-way analysis of variance (ANOVA). The data is presented as a mean standard deviation. When *p* < 0.05, differences were significant. All of the experiments were performed three times.

## 3. Results and Discussion

### 3.1. Analysis of Results of Single Factor Test

According to the data in Figure 1a, the extraction rate of cotton flowers flavonoids was the highest when the extraction time was 1.5 h, and the extraction rate decreased when the extraction time continued to extend. Therefore, the 1.5 h extraction time was chosen as the best extraction time.

Figure 1b shows that when the liquid-solid ratio was 1:20, cotton flowers have the highest rate of flavonoid extraction, and the extraction rate dropped as the liquid-solid ratio continued to rise. Therefore, the optimal liquid-solid ratio was 1:20.

Figure 1c shows that the extraction rate of flavonoids in cotton flowers was the highest when the extraction temperature was 50 °C, and the extraction rate decreased with the increasing temperature. Therefore, 50 °C was selected as the optimal extraction temperature.

As shown in Figure 1d, the extraction rate of cotton flowers flavonoids was the highest when the ethanol concentration was 70%, and the extraction rate decreased when the ethanol concentration continued to increase. Therefore, 70% ethanol was chosen as the optimal extraction concentration.

The final results can be obtained by analyzing the data of these single factors methods. The optimal extraction process conditions were as followed: extraction time: 1.5 h; extraction temperature: 50 °C; liquid-solid ratio: 1:20; ethanol concentration: 70%. Then we designed a RSM experiment to determine the optimal extraction parameters and improve the extraction yield and eliminate the wastage between the single factors.

### 3.2. Data Identification by RSM

According to the Box-Behnken principle, an experimental program was designed to investigate the effects of four factors: extraction time, liquid-solid ratio, extraction temperature, and ethanol concentration on the extraction rate of cotton flowers flavonoids. A total of 29 experimental points were designed for the RSM test. The design table and results were shown in Table 1. The influence of two interaction factors on the light absorption value of flavonoids in cotton flowers was observed and other factors were fixed to zero.

The results in Table 2 showed the influence of two interaction factors on the light absorption value of flavonoids in cotton flowers by RSM, and other factors were fixed at zero. According to the regression equation, the shape of the response surface of each factor was investigated, and the influence of each factor on the absorption value of flavonoids in cotton flowers was analyzed. The *p-*value of the sample in Table 2 is 0.0135 (*p* < 0.05), which is significant, indicating that the model is fitting. The Lack of Fit data is 0.1779, which is proven meaningful.

### 3.3. Response Surface Analysis

Design-Expert.V8.0.6 software (Minneapolis, MN, USA) was used for data analysis, and a three-dimensional effect surface diagram was generated, with the interaction between various factors shown in Figure 2. Figure 2a shows the impact of extraction duration and liquid-solid ratio on flavonoids extraction rate. The temperature of the extraction was kept constant at 50 °C, the ethanol concentration at 70%, a significant interaction between the time of extraction and the liquid-solid ratio. The extraction rate increased as the liquid-solid ratio increased from 15 to 24.63, and the extraction time increased from 1 h to 2 h. However, when the liquid-solid ratio exceeded 24.63, the extraction rate began to decrease.

From Figure 2b, it can be seen that the effect of extraction time and temperature on flavonoids extraction yield. The liquid-solid ratio was kept at 20, while the ethanol concentration was at 70%. Figure 2b shows a significant interaction between extraction time and extraction temperature. When the time of extraction was from 1 h to 2 h, the extraction temperature from 45 °C to 53.35 °C the extraction rate of flavonoids increased. When the extraction temperature increased from 53.35 °C to 55 °C, the rate of flavonoid extraction decreased.

The effects of extraction time and ethanol concentration on the flavonoid extraction rate were found in Figure 2c. When the liquid-solid ratio was kept at 20, the extraction temperature was 50 °C. Figure 2c shows that extraction time has significant interaction with ethanol concentration. The time increased from 1 h to 2 h, the ethanol concentration increased from 65% to 75%, and the extraction rate of flavonoids continued to increase.

Figure 2d shows the effects of extraction temperature and liquid-solid ratio on flavonoids extraction rate. The extraction time was kept at 1.5 h, and the ethanol concentration at 70%. In Figure 2d, the interplay between extraction temperature and the liquid-solid ratio was found to be substantial, when the liquid-solid ratio was increased from 15 to 24.63, the extraction temperature was increased from 45 °C to 53.35 °C. The extraction rate was also improved. When the liquid-solid ratio was increased from 24.63 to 25, the extraction time was extended from 1 to 2 h. When the liquid-solid ratio exceeded 24.63, the extraction temperature increased from 53.35 °C to 55 °C, and the rate of flavonoids extraction reduced.

Figure 2e demonstrates the impact of the liquid-solid ratio and ethanol concentration on flavonoids extraction yield. When the time of extraction was kept at 1.5 h, the extraction temperature was kept at 50 °C, Figure 2e showed no significant interaction between solid-liquid ratio and ethanol content.

Figure 2f shows the effects of extraction temperature and ethanol concentration on flavonoids extraction rate. The liquid-solid ratio was kept at 20, and the time of extraction was at 1.5 h. Figure 2f showed that extraction temperature has significant interaction with ethanol concentration. The extraction temperature increased from 45 °C to 53.35 °C, the ethanol concentration then increased from 65% to 75%, and the extraction rate of flavonoids increased continuously.

The significance of each model coefficient was assessed using Figure 2g *t*-tests. The best conditions for the extraction process of flavonoids were extraction time at 2 h, and an extraction temperature of 53.35 °C. The liquid-solid ratio was 24.63, and the ethanol concentration was 75%. Under these conditions, the flavonoid yield was 5.66%.

### 3.4. Antioxidant Activity Test Analysis

As shown in Figure 3a, at 0.025–0.1 mg/mL, the activity of flavonoids in DPPH free radical scavenging was inferior to that of BHT. When there is a concentration of 0.025 mg/mL, the free radical scavenging activity of DPPH of flavonoids obtained by the DT method was slightly better than that of LNT. When the concentration reached 0.1 mg/mL, the activity of flavonoids to scavenge DPPH free radicals increased significantly and reached the level of BHT. With the increase in the concentration of flavonoids, the activity of scavenging DPPH was also increased.

From Figure 3b, it can be seen that in the concentration range of 0.025–0.4 mg/mL, BHT’s ability to scavenge hydroxyl radicals increases as its concentration rises. When the concentration reaches 0.4 mg/mL, the scavenging rate has reached 99.8%. At 0.025–0.4 mg/mL concentration, the scavenging activity of LNT flavonoids enhanced when the concentration was raised, and the clearance rate was between 40% and 60%. The hydroxyl radical scavenging activity of DT flavonoids also increased with the increase in concentration, and the scavenging rate was slightly higher than that of LNT, ranging from 48% to 60%. These results indicate that flavonoids have a particular ability to scavenge hydroxyl radicals. The ability of LNT flavonoids and DT flavonoids to scavenge hydroxyl radicals in the concentration range of 0.025–0.4 mg/mL was lower than that of BHT.

Figure 3c demonstrates that in the concentration range of 0.025–0.4 mg/mL, LNT flavonoids and DT flavonoids scavenging superoxide anion radical scavenging activities are higher than BHT, which indicates that flavonoids have strong antioxidant capacity. Within the concentration range established by the experiment, the antioxidant activity increased with increasing concentration.

Figure 3d shows that, in the concentration range used in the experiment, the reducing ability of BHT to Fe^3+^ is higher than that of flavonoids, and the reducing ability of DT, LNT, and BHT all increase with the increase of concentration, showing a concentration dependence.

### 3.5. Cell Viability Results

As can be seen from Figure 4, the blank group’s cell survival rate was greater, while the survival rate of HT-22 cells with glutamate-induced injury was about 60%. The survival rate of HT-22 cells treated with cotton flower flavonoids of 1 µg/mL, 5 µg/mL, 25 µg/mL, and 100 µg/mL was higher than that of HT-22 cells without flavonoids. At low concentrations (1 µg/mL and 5 µg/mL), the protective effect was not very obvious. When the concentration was raised to 100 µg/mL, the cell survival rate was 94%. These results indicated that flavonoids from cotton flowers could protect glutamate-induced HT-22 cells, and the higher the concentration, the better the protective effect.

## 4. Discussion

The main extraction methods of flavonoids include organic solvent extraction, microwave extraction, and water extraction [26,27]. The process flow of these extraction methods is very complicated, with a long extraction time, high energy consumption, insufficient extraction of active ingredients, significant loss of organic solvent, and low extraction rate.

In order to overcome the shortcomings of the above existing technologies and improve the yield of flavonoids, the current extraction technology is mainly applied to these plants such as Xinjiang jujube [11], navel orange [12], and rape bee pollen [13]. The advantages of RSM are high extraction rate, short time, low energy consumption, and precise parameter control [28]. Compared with other extraction processes, such as microwave extraction, organic solvent extraction, hot water extraction, and ammonia calcium chloride extraction, RSM is more suitable for extracting cotton flowers flavonoids. According to the Box-Behnken principle, the influence of four factors, including extraction time, liquid-solid ratio, extraction temperature, and ethanol concentration, on the extraction rate of cotton flowers flavonoids was investigated by RSM.

The conditions for extracting flavonoids from cotton blossoms had been tested by using RSM [29]. Their three-factor result showed that the extraction temperature was 180 °C with a 65% ethanol concentration and a liquid-solid ratio of 65 mL/g. Our current work is to optimize the RSM conditions for reflux extraction, and what we do are four factors and three levels. Moreover, the extraction temperature used by Xu is 180 °C whereas ours is 53.4 °C, which significantly increases the reproducibility of the extraction process and reduces the cost of industrial production.

In the antioxidant experiments, the antioxidant results of our group were similar to those of Chetan [30], Zvezdelina [31], and Dinh [32]. The results of the DPPH scavenging test for free radicals, superoxide anion free radical scavenging test, and high-concentration partial reducing ability test showed that LNT flavonoid powder has a higher free radical scavenging ability, which may be due to the low temperature protecting the structure of flavonoids from antioxidant activity.

The hydroxyl radical scavenging test revealed that flavonoids’ hydroxyl radical scavenging ability was almost half of BHT. It is possible that the experimental system is susceptible to the influence of ions in water, so the result of flavonoids from cotton flowers suggested a certain scavenging ability.

The protective effect of flavonoids on cells has not been reported, and the results of our research group show that flavonoids can protect HT22 cells induced by glutamate. Our work increases the yield of flavonoids extracted from waste cotton, further broadens the utilization value of cotton, and proves that cotton extract flavonoids are a potent antioxidant that can be further used for medicinal purposes.

## Figures and Tables

**Figure 1 molecules-27-05627-f001:**
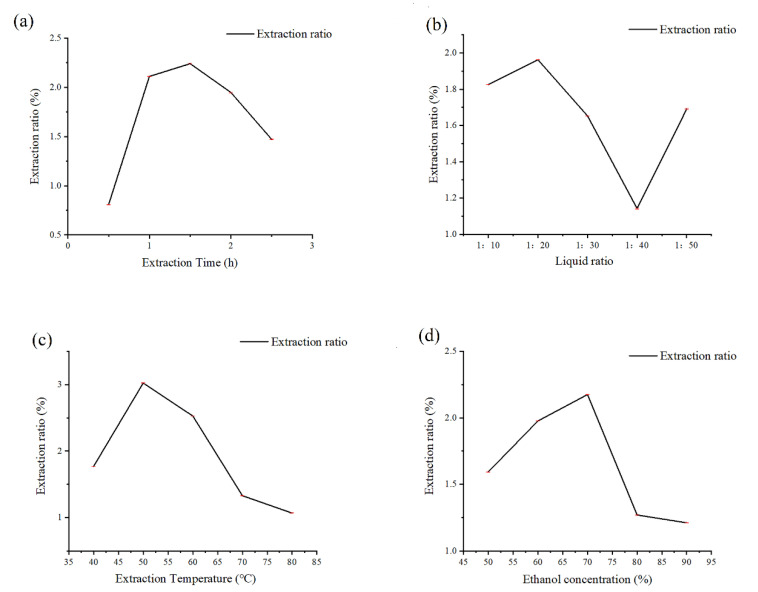
Analysis of single factor test results of extraction time, liquid ratio, extraction temperature, and ethanol concentration. (**a**): Effect of extraction time on the extraction rate of flavonoids from cotton flowers; (**b**): Influence of ratio of liquid to material on extraction rate of flavonoids from cotton flowers; (**c**): Influence of extraction temperature on extraction rate of flavonoids from cotton flowers; (**d**): Effect of ethanol concentration on extraction rate of flavonoids from cotton flowers.

**Figure 2 molecules-27-05627-f002:**
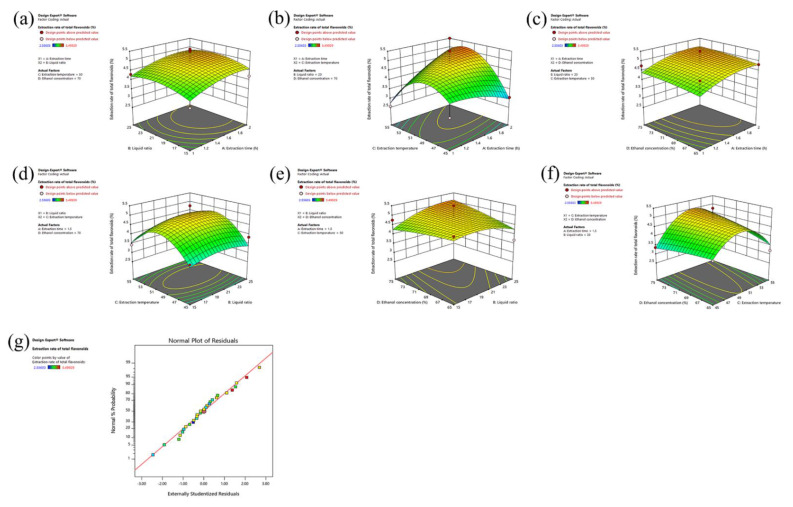
Response Surface Methodology Interaction of Four Factors Affecting the Extraction Rate of Flavonoids from Cotton Flowers. (**a**): The effect of the interaction between liquid-solid ratio and extraction time on the yield of flavonoids. (**b**): The effect of the interaction of extraction temperature and time on the yield of flavonoids. (**c**): The effect of the interaction between ethanol concentration and extraction time on the yield of flavonoids. (**d**): The effect of the interaction between ethanol concentration and liquid-solid ratio on the yield of flavonoids. (**e**): The effect of the interaction between ethanol concentration and extraction temperature on the yield of flavonoids. (**f**): The effect of the interaction between ethanol concentration and extraction temperature on the yield of flavonoids. (**g**): Normal distribution.

**Figure 3 molecules-27-05627-f003:**
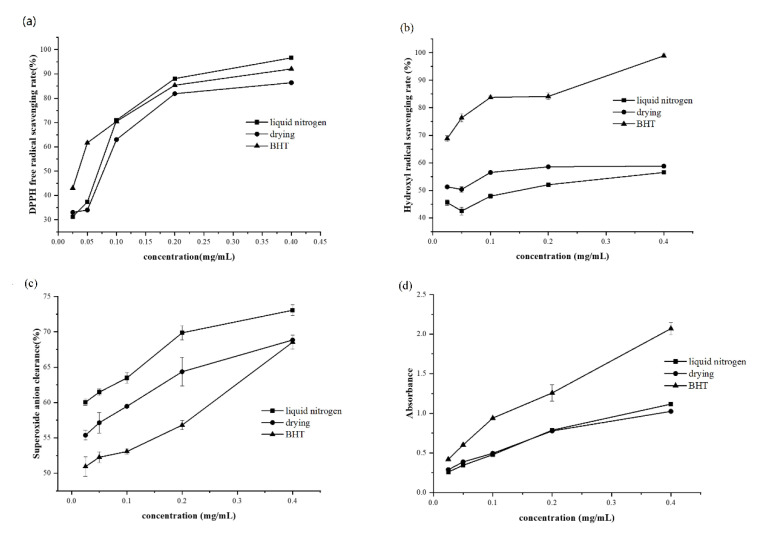
Antioxidant activity test result. (**a**): DPPH free radical scavenging test results. (**b**): Hydroxyl radical scavenging test results. (**c**): Results of superoxide anion radical scavenging test. (**d**): Fe^3+^ reduction experiments.

**Figure 4 molecules-27-05627-f004:**
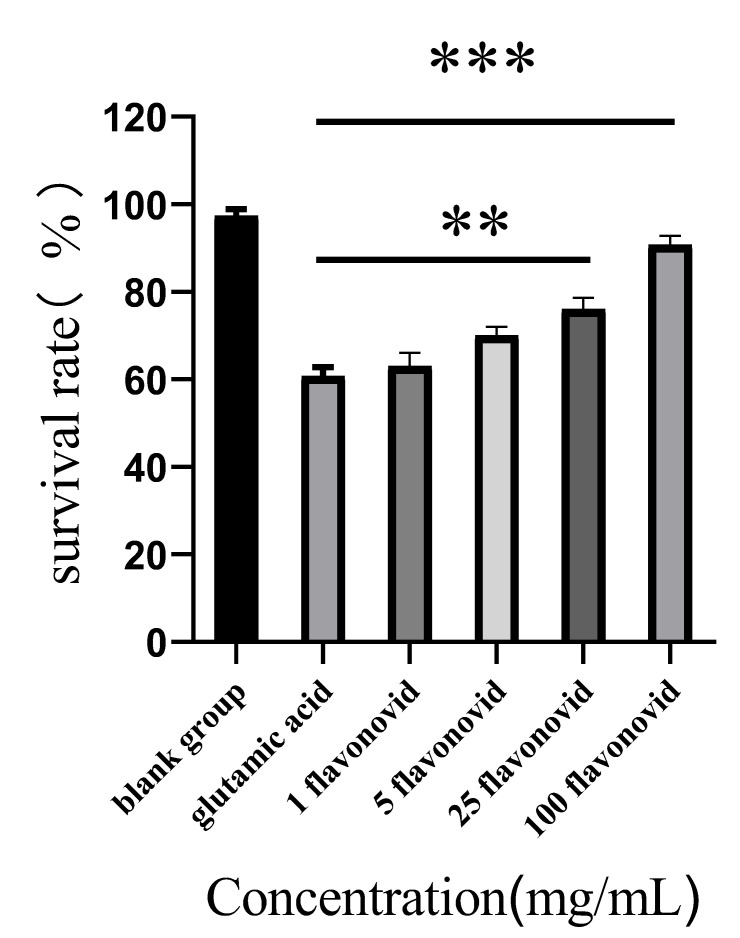
The protective effect of flavonoids on HT-22 cells (** *p <* 0.01, *** *p <* 0.0001).

**Table 1 molecules-27-05627-t001:** Design Results of Boxed Behnken Center combined experiment for cotton flowers.

Test Number	Time	Liquid Ratio	Temperature	Ethanol Concentration	Extraction Rate of Flavonoids
1	−1	−1	0	0	3.62695
2	1	−1	0	0	4.208511
3	−1	1	0	0	4.293617
4	1	1	0	0	4.875177
5	0	0	−1	−1	3.761702
6	0	0	1	−1	3.208511
7	0	0	−1	1	3.371631
8	0	0	1	1	4.201418
9	−1	0	0	−1	4.91773
10	1	0	0	−1	4.811348
11	−1	0	0	1	4.747518
12	1	0	0	1	4.655319
13	0	−1	−1	0	3.513475
14	0	1	−1	0	3.846809
15	0	−1	1	0	3.428369
16	0	1	1	0	4.222695
17	−1	0	−1	0	3.151773
18	1	0	−1	0	3.052482
19	−1	0	1	0	2.556028
20	1	0	1	0	5.499291
21	0	−1	0	−1	4.860993
22	0	1	0	−1	3.69078
23	0	−1	0	1	4.75461
24	0	1	0	1	4.797163
25	0	0	0	0	4.740426
26	0	0	0	0	5.499291
27	0	0	0	0	4.81844
28	0	0	0	0	4.733333
29	0	0	0	0	4.662411

**Table 2 molecules-27-05627-t002:** Variance analysis table of flavonoid absorbance of cotton flowers optimized by Response Surface Method.

Source	Sum of Squares	df	Mean Square	F-Value	*p*-Value
Model	12.60	14	0.9002	3.45	0.0135 significant
A-Extraction time	1.21	1	1.21	4.64	0.0492
B-Liquid ratio	0.1481	1	0.1481	0.5682	0.4635
C-Extraction temperature	0.4874	1	0.4874	1.87	0.1931
D-ethanol concentration	0.1358	1	0.1358	0.5209	0.4823
AB	2.505 × 10^−13^	1	2.505 × 10^−13^	9.606 × 10^−13^	1.0000
AC	2.31	1	2.31	8.88	0.0100
AD	0.0001	1	0.0001	0.0002	0.9891
BC	0.0531	1	0.0531	0.2038	0.6586
BD	0.3677	1	0.3677	1.41	0.2548
CD	0.4782	1	0.4782	1.83	0.1971
A^2^	0.3476	1	0.3476	1.33	0.2675
B^2^	0.45492	1	0.4592	1.76	0.2057
C^2^	7.21	1	7.21	27.65	0.0001
D^2^	0.0223	1	0.0223	0.0855	0.7742
Residual	3.65	14	0.2607		
Lack of Fit	3.18	10	0.3175	2.67	0.1779 not significant
Pure Error	0.4751	4	0.1188		
Cor Total	16.25	28			

## Data Availability

Not applicable.

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
