# Peer review of "Effects of Extraction Technique on the Content and Antioxidant Activity of Flavonoids from Gossypium *Hirsutum linn*. Flowers"

_molecules, 2022, doi:10.3390/molecules27175627_

Round 1
Reviewer 1 Report
Authors have performed extensive work to improve the flavonoids extraction procedure of cotton flowers in China. Results from the study can be employed in agriculture to increase the flavonoids yield. All sections of the manuscript are well written.
Suggestions:
p4/line 143 - Please, write dulbecco's with "D"
p4/l 144 and 145 - Please, provide the producer of DMEM and antibiotic supplement, respectively
p12/l 365 - The same for dulbecco's
In Discussion section or in Introduction - please, include the following paper emphasising the extraction procedure, by Yaneva, Z. et al. 2022. Separations, 9(2), 27 . Flavonoids extraction kinetics......
Author Response
Q1: p4/line 143 - Please, write dulbecco's with "D"
A: Done.
Q2: p4/l 144 and 145 - Please, provide the producer of DMEM and antibiotic supplement, respectively
A: Done.
Q3: p12/l 365 - The same for dulbecco's
A: Done.
Q4:in Discussion section or in the Introduction - please, include the following paper emphasizing the extraction procedure, by Yaneva, Z. et al. 2022. Separations, 9(2), 27. Flavonoids extraction kinetics......
A: Done. (Line 355 in the revised manuscript)
Reviewer 2 Report
They suggested that extraction of cotton flowers flavonoids have antioxidant acitivity.
But I think there is no exact method of separating flavonoids from cotton in manusicirit and the results are very poor. It is not known what the result value the author is suggesting, and this reviewer has no way to confirm how much flavonoids will be contained in the extract as the result the author presents.
It cannot be concluded that it is simply antioxidant activity through the cell MTT test results of the substance measured for xanthine oxide activity.
This reviewer believes that this paper cannot be published in Molecules.
1. What function of flavonoids do you think work between cotton and other plants in line 55?
2. Line 71-83, how can you sure that power is flavonoids.
3. There is no explanation for the experimental process cell culture.
4. They treated flavonoid or 24, 48, or 72 h in Materials and methods, but result show only 1 incubation time.
5. Every figures should be statistics.
Reviewer 3 Report
This study analyzed how to better extract flavonoids from cotton, which is a very good exploration. Cotton, a widely grown crop, has a high yield and is relatively simple to obtain. After extracting the flavonoids, the authors conducted antioxidant capacity tests as well as cell damage protection experiments, and the results showed that cotton flavonoids have good applicability.
However, the following aspects need to be improved in this paper.
1. in Table 1, the data of assay 20 and 26 are consistent, which indicates that the optimized result is not better than the result before optimization.
2. in Table 1, data on the effect of single factor variation on extraction rate need to be added
3. In Table 1, the data variation of 25-29 fluctuates a lot
4. the results of Figure2 were fitted, were multiple extraction repetitions performed?
5. How was the BHT done? Or what is the positive control group?
6. The legend of Figure 4 is not very clear and does not give the reader a clear picture of the sample treatment.
7. Figure 4 needs to be analyzed for significance of differences
Round 2
Reviewer 2 Report
your revision is improved thab last version. but i dont agree publishing